# MixQuant: A Quantization Bit-Width Search that can Optimize the Performance of Your Quantization Method

## Abstract

Quantization is a technique for creating efficient Deep Neural Networks (DNNs), which involves performing computations and storing tensors at lower bit-widths than f32 floating point precision. Quantization reduces model size and inference latency, and therefore allows for DNNs to be deployed on platforms with constrained computational resources and real-time systems. However, quantization can lead to numerical instability caused by roundoff error which leads to inaccurate computations and therefore, a decrease in quantized model accuracy. In this paper we focus on simulated quantized inference, where the quantized model parameters are stored in low-precision, but the mathematical operations on them (e.g. matrix multiplications and additions) are performed with floating point arithmetic. This means that the DNN parameters are first quantized from f32 to, for example, int4, and then dequantized back to f32 to perform computations. We show that the roundtrip process of quantizing and dequantizing the model parameters leads to roundoff error, which may lead to numerical instability. Similarly to prior works, which have shown that both biases and activations are more sensitive to quantization and are best kept in full precision or quantized with higher bit-widths, we show that some weights are more sensitive than others which should be reflected on their quantization bit-width. To that end we propose *MixQuant*, a search algorithm that finds the optimal custom quantization bit-width for each layer weight based on roundoff error and can be combined with any quantization method as a form of pre-processing optimization. We show that combining *MixQuant* with BRECQ, a state-of-the-art quantization method, yields better quantized model accuracy than BRECQ alone. Additionally, we combine *MixQuant* with vanilla asymmetric quantization to show that *MixQuant* has the potential to optimize the performance of any quantization technique.

## 1 Introduction

Quantization is a method for mapping continuous values to a set of discrete values. The goal of neural network quantization is to perform computations and store tensors at lower bit-widths than floating point precision to reduce model size and inference latency while maintaining model accuracy, which allows for deploying DNNs on platforms with constrained computational resources, e.g.: real time inference on mobile devices. Quantization can be performed during training or inference. In this paper we focus on quantized inference, specifically post-training quantization, which quantizes a full precision trained model without the need for re-training or fine-tuning.

Quantized inference can be either simulated or integer-only, and in this paper we focus on simulated quantization, where the quantized model parameters are stored in low-precision, but the mathematical operations on them (e.g. matrix multiplications and additions) are performed with floating point arithmetic (Gholami et al., 2022). In Tensorflow, PyTorch, and HuggingFace (QDQBERT model), simulated quantization is referred to as fake quantization. This means that the DNN parameters are first quantized from f32 to, for example, int4, and then dequantized back to f32 to perform the forward pass executed during inference. We show that the roundtrip process of quantizing and dequantizing the model parameters leads to roundoff error, which may lead to numerical instability.

Similarly to prior works, which have shown that both biases and activations are more sensitive to quantization and are best kept in full precision or quantized with higher bit-widths (Zhou et al., 2016), we show that some weights are more sensitive than others which should be reflected on their quantization bit-width. To that end we propose *MixQuant*, a search algorithm that finds the optimal quantization bit-width from int2, int3, int4, int5, int6, int7, and int8 for each layer weight based on roundoff error and can be combined with any quantization method as a form of pre-processing optimization. We show that combining *MixQuant* with BRECQ (Li et al., 2021), a state-of-the-art quantization method, yields better quantized model accuracy than BRECQ alone. Additionally, we combine *MixQuant* with vanilla asymmetric quantization to show that *MixQuant* has the potential to optimize the performance of any quantization technique.

*MixQuant* has three main benefits. First, *MixQuant* is a component of the quantization process, which can be leveraged to find optimal quantization mixed precision bit-widths that can be plugged into any quantization method to optimize its performance. Second, *MixQuant* is linear and runs in a matter of seconds, which makes it practical. Third, combining *MixQuant* with BRECQ, a state-of-the-art quantization method yields better quantized model accuracy than BRECQ alone, OMSE (Choukroun et al., 2019), AdaRound (Nagel et al., 2020), AdaQuant (Hubara et al., 2020), and Bit-Split (Wang et al., 2020).

## 2    RELATED WORK

**Neural Network Quantization**    Neural network quantization can be applied to training (Gupta et al., 2015; Zhou et al., 2016; Hubara et al., 2017; Bartan & Pilanci, 2021; Elthakeb et al., 2020) or inference. There are two paradigms in quantized DNN inference: post-training quantization (PTQ) and quantization-aware training (QAT) (Jacob et al., 2018; Tailor et al., 2021). In contrast to PTQ, QAT requires that the f32 model is retrained while simulating quantized inference in the forward pass. While *MixQuant* can be integrated with either, we focus on PTQ which does not require any re-training.

Hubara et al. (2021) and Li et al. (2021) are amongst the current state-of-the-art post training quantization works. Hubara et al. (2021) introduce AdaQuant, which finds optimal quantization for both weights and activations and is based on minimizing the error between quantized layer outputs and f32 layer outputs. This approach is similar to *MixQuant*; however, *MixQuant* finds the optimal quantization bit-widths based on quantization error (QE) minimization, while AdaQuant treats the bit-width as a constant and quantizes all weights and activations using the same bit-width (either *int8* or *int4*). Li et al. (2021) propose BRECQ, a quantization method based on DNN block reconstruction. Nagel et al. (2020) propose AdaRound, adaptive rounding for weights, which achieves better accuracy than rounding to the nearest. They formulate the rounding procedure as an optimization problem that minimizes the expected difference between model loss with and without weights quantization perturbation. Li et al. (2020) develop a method based on constraining all quantization levels as the sum of Powers-of-Two terms, Wang et al. (2020) propose a Bit-Split and Stitching framework (Bit-split), Nahshan et al. (2021) study the effect of quantization on the structure of the loss landscape, Banner et al. (2019) develop ACIQ-Mix, a 4 bit convolutional neural network quantization, and Cai et al. (2020) perform zero-shot quantization ZeroQ based on distilling a dataset that matches the input data distribution.

Quantization originated with convolutional neural networks, but it has been extended to natural language processing neural networks as well. Chen & Sun (2020) propose differentiable product quantization, a learnable compression for embedding layers in DNNs. Kim et al. (2021) study an integer-only quantization scheme for transformers, where the entire inference is performed with pure integer arithmetic.

Other works studied hardware optimization for quantization or the relationship between quantization and adversarial robustness. Han et al. (2020) focus on performance optimization for Low-bit Convolution on ARM CPU and NVIDIA GPU. Fu et al. (2021) investigate quantized models' adversarial robustness. They find that when an adversarially trained model is quantized to different precisions in a post-training manner, the associated adversarial attacks transfer poorly between different precisions.

**Mixed Precision Quantization** In this paper we focus on mixed precision quantization. There are only a few prior works that focus on mixed precision quantization since most focus on single precision quantization, where the quantization bit-width of all weights are uniform and therefore; treated as a constant. Wang et al. (2019) propose a framework for determining the quantization policy with mixed precision and reinforcement learning, but compared to *MixQuant* it requires significantly more overhead (hardware simulators and reinforcement learning). Liang (2020) focuses on mixed precision quantization of activations and distinguishes between key and non-key activations to assign 8-bit and 4-bit precision respectively. In contrast to MixQuant, which searches for weights mixed precision from 8 to 2 bits, Liang (2020) is limited to a choice between 4 and 8 bits and applies only to activations while all weights are quantized with 8-bit precision. The primary focus of Wu et al. (2018) is neural architecture search, which can also be used for mixed precision quantization. However, their search on ResNet 18 for ImageNet takes 5 hours, while *MixQuant* runs in order of a few seconds. Liu et al. (2021) use single precision for weights, where the mixed precision is represented only by selecting a different bit-width for weights than activations. Liu et al. (2021) is the most most recent, and we show that *MixQuant* yields better accuracy.

Another mixed precision quantization work that we build on is Lin et al. (2016), who identify optimal bit-width allocation across DNN layers. However, there are two primary differences between Lin et al. (2016) and our work: (1) Lin et al. (2016) focus on fixed-point precision, not integer precision, (2) Lin et al. (2016) a different method for finding layer bit-widths based on predicted signal-quantization-to-noise -ratio. Moreover, while they find that on CIFAR-10 convolutional DNN is able to achieve 20 % model size reduction; their AlexNet experiments on ImageNet-1000 achieve less than 1% model reduction. In this work we are able to successfully leverage mixed precision optimal bit-width allocation on ImageNet-1000 models.

## 3 QUANTIZATION AND NUMERICAL INSTABILITY

Quantization involves lowering the bit-width of a numeric tensor representation, which can cause numerical instability that leads to inaccurate outputs (Kloberdanz et al., 2022). In general, numerical instability arises due to two types of numerical errors: (1) roundoff errors and (2) truncation errors. Roundoff errors are caused by approximating real numbers with finite precision, while truncation errors are caused by approximating an iterative mathematical process with only a finite number of iterations. We argue that quantization can significantly amplify the roundoff error, which leads to a degradation in quantized DNN accuracy.

DNN training and inference is typically performed in f32 precison, which already introduces roundoff errors, because it has only 32 bits to represent real numbers. Specifically, f32 can represent a zero and numbers from -3.40282347E+38 to -1.17549435E-38 and from 1.17549435E-38 to 3.40282347E+38, but numbers outside of this range are not representable in f32. In simulated quantization the process of quantizing DNN parameters from f32 to int (e.g.: int4) and dequantizing them back to f32 to perform matrix multiply and add (e.g.: inputs * weights + biases) can lead to a loss of precision.

Listing 1 shows an example of a simple simulated quantized inference, where the weights tensor is quantized to int2 and its subsequent dequantization back to f32 has a roundoff error. The roundoff error occurs in the second element of the weight tensor, which becomes 0.0 (line 40) while its true original value is 0.01 (line 38). This error caused by quantization then propagates further - the computation $inputs * weights + biases$ returns 1.0000e-05 (line 43) instead of 1.2000e-05 (line 42) in the second element of the result tensor.

Listing 1: Loss of Precision due to Quantization Example

```
1
2   def scale(r, bits):
3       min_r = r.min()
4       max_r = r.max()
5       qmin = -1 * (2 ** (bits - 1))
6       qmax = 2 ** (bits - 1) - 1
7       scale_r = (max_r - min_r) / (qmax - qmin)
8       return scale_r
9
```

```
10   def zero_point(r, bits):
11       scale_r = scale(r, bits)
12       min_r = r.min()
13       qmin = -1 * (2 ** (bits - 1))
14       zpt_r = qmin - int(min_r / scale_r)
15       return zpt_r
16
17   def quant(r, bits):
18       z = zero_point(r, bits)
19       s = scale(r, bits)
20       q = (torch.round(r/s) + z).int()
21       return q
22
23   def dequant(q, z, s):
24       r = (q - z) * s
25       return r.float()
26
27   input = torch.tensor([0.005, 0.0002, 0.01, 0.003])
28   bias = torch.tensor([0.00001])
29   weight = torch.tensor([-1.0, 0.01, 1.0, 2.0]) # original weight
30
31   S = scale(weight, 2) # quantization scale
32   Z = zero_point(weight, 2) # quantization zero point
33   q_weight = quant(weight, 2) # quantized weight
34   dq_weight = dequant(q, Z, S) # dequantized weight
35
36   result = input * weight + bias
37   dq_result = input * dq_weight + bias
38
39   f32 weight:  tensor([-1.0000,  0.0100,  1.0000,  2.0000])
40   quantized weight:  tensor([-2, -1,  0,  1], dtype=torch.int32)
41   dequantized weight:  tensor([-1.,  0.,  1.,  2.])
42
43   f32 result:  tensor([-4.9900e-03,  1.2000e-05,  1.0010e-02,  6.0100e-03])
44   simulated quantization result:  tensor([-4.9900e-03,  1.0000e-05,  1.0010e-02,
     6.0100e-03])
```

## 4 MIXQUANT

*MixQuant* is a quantization scheme that relies on mixed precision to find the bit-widths of individual layer weights that minimize roundoff error and therefore, minimize model accuracy degradation due to quantization. Specifically, *MixQuant* is a search algorithm that finds optimal bit-widths that minimize model accuracy degradation caused by quantization. Prior works have shown that biases and activations are more sensitive to quantization than weights, and are therefore typically kept in higher precision. In this paper we argue some weights are more sensitive to quantization than others, which we show in our ablation studies. This warrants a careful bit-width allocation to individual weights and serves as motivation for MixQuant. In essence, *MixQuant* can be viewed as an additional pre-processing optimization component of the quantization process, which can be combined with any quantization method optimize its performance.

*MixQuant* is described in Algorithm 1. The optimal weight layer bit-widths search has two primary components: layer-wise QE minimization and a QE multiplier (QEM). The layer-wise QE is calculated as the mean squared error (MSE) between the f32 model weights and the weights that have been dequantized following an int quantization (any quantization method can be used at line 8 in Algorithm 1) to capture the information loss due to roundoff error caused by quantization. This error is calculated for each layer for each bit-width from the following list: 8, 7, 6, 5, 4, 3, and 2 (lines 4-11 in Algorithm 1). Following that, *MixQuant* searches for the optimal bit-width for each layer by comparing the QE of each bit-width from this list with an int8 error, which serves as a baseline (lines 12-13 in Algorithm 1). To push *MixQuant* to select bit-widths lower than int8, *MixQuant* leverages the QEM. If the QE at a bit-width b is less or equal to int8 QE multiplied by the QEM, b becomes

the optimal bit-width for that layer. This can be expressed as an optimization problem:

$$optBit = \arg\min_{optBit} quantErrors$$
$$\text{Subject to } quantErrors \leq 8bit_qError * QEM \quad (1)$$
$$optBit \in B$$

Because the QEM is an input parameter into MixQuant, it allows the user to specify a custom trade-off between quantization bit-width and model accuracy; and therefore, it allows the user to find *their* optimal layer bit-width.

---

**Algorithm 1** MixQuant

1: **Input:** full precision weights $W$, bit-widths $B$, QE multiplier $QEM$
2: Initialize $optimalBitWidths$
   /* Iterate over all layers */
3: **for** $l$ **in** $layers$ **do**
4:     $8bit_W$ = Quantize($W$, $bitWidth = 8$)
       /* Compute int8 quantization error in layer l */
5:     $8bit_qError$ = $W$ - Dequantize($8bit_W$)
       /* For every bit-width in $B$ compute quantization error in layer l */
6:     Initialize $quantErrors$
7:     **for** $bitWidth$ **in** $B$ **do**
8:         $quantizedW$ = Quantize($W$, $bitWidth$)
9:         $qError$ = $W$ - Dequantize($quantizedW$)
10:        Append $qError$ to $quantErrors$
11:    **end for**
       /* Select optimal bit-width at layer l */
12:    $optBit$ = $\arg\min_{optBit}$ quantErrors  s.t.
                 quantErrors $\leq$ $8bit_qError$ * $QEM$,
                 $optBit \in B$
13:    Append $optBit$ to $optimalBitWidths$
14: **end for**
15: **return** $optimalBitWidths$

---

**Weights Mixed Precision Quantization**   We focus on weights quantization for three reasons. First, weights account for majority of parameters in a DNN and therefore, have the greatest impact on model size and inference time. Second, model accuracy is more sensitive to quantized activations than weights (Zhou et al., 2016). Third, we guided our algorithm design with the state-of-the art results in table 2 in Li et al. (2021), who introduced BRECQ which shows weight-only quantization.

**Approximating Roundoff Error**   We use the QE (measured as the MSE between f32 and dequantized weights) to approximate the impact of quantization on model accuracy for three reasons. First, prior works have leveraged quantization error as a proxy for quantized model accuracy - Banner et al. (2019) used quantization MSE to approximate optimal clipping value (ACIQ) and optimal bit-width for each channel. Second, we provide empirical evidence that there is a negative relationship between model accuracy and quantization error (see Figure 6 in Appendix). Third, computing layer-wise quantization error instead of determining the model accuracy with respect to each layer and each possible layer bit-width has the advantage of linear time complexity. An exhaustive combinatorial search runs in exponential time (Wu et al., 2018).

**Time Complexity Analysis**   We analyze the algorithm's time complexity by considering its two logical components - the error calculations and the bit-width search based on them. Let L be the total number of layers, B the total number of bit-widths, and M the total of QEMs. We calculate the QE of each layer for each bit-width. Thus, the time complexity of *MixQuant's* error calculations (line 4-11 in Algorithm 1) is $\mathcal{O}(L * B)$. The bit-width search (line 12-13 in Algorithm 1) compares the QE of each bit-width to the baseline int8 QE for each layer and can be performed for M number of QEMs, which takes $(L * (B * M))$.

Table 1: Model accuracy comparison of *MixQuant* combined with BRECQ and BRECQ alone

| | Bits W/A | *MixQuant* + BRECQ | **f32 vs *MixQuant* + BRECQ** | Bits W/A | BRECQ | f32 vs BRECQ |
|---|---|---|---|---|---|---|
| ResNet-18 | 32/32 | 69.76 | | 32/32 | 71.08 | |
| | 4, 5, 6/32 | 70.69 | **0.93** | 4/32 | 70.7 | -0.38 |
| | 4, 5, 6/32 | 70.69 | **0.93** | 3/32 | 69.81 | -1.27 |
| | 2, 5, 6/32 | 68.93 | **-0.83** | 2/32 | 66.3 | -4.78 |
| MobileNetV2 | 32/32 | 71.88 | | 32/32 | 72.49 | |
| | 4, 5, 6, 7/32 | 71.92 | **0.04** | 4/32 | 71.66 | -0.83 |
| | 4, 5, 6, 7/32 | 71.92 | **0.04** | 3/32 | 69.5 | -2.99 |
| | 2, 5, 6/32 | 59.53 | **-12.35** | 2/32 | 59.67 | -12.82 |

Table 2: Model accuracy comparison of *MixQuant* combined with BRECQ and Liu et al. (2021)

| | Bits W/A | *MixQuant* + BRECQ | **f32 vs *MixQuant* + BRECQ** | Bits W/A | Liu et al. (2021) | f32 vs Liu et al. (2021) |
|---|---|---|---|---|---|---|
| ResNet-18 | 32/32 | 69.76 | | 32/32 | 74.24 | |
| | 4, 5, 6/32 | 70.69 | **0.93** | 4/8 | 61.68 | -12.56 |
| | 4, 5, 6/32 | 70.69 | **0.93** | 4/8 | 61.68 | -12.56 |
| | 2, 5, 6/32 | 68.93 | **-0.83** | 4/8 | 61.68 | -12.56 |
| MobileNetV2 | 32/32 | 71.88 | | | 71.78 | |
| | 4, 5, 6, 7/32 | 71.92 | **0.04** | 8/8 | 70.7 | -3.54 |
| | 4, 5, 6, 7/32 | 71.92 | **0.04** | 8/8 | 70.7 | -3.54 |
| | 2, 5, 6/32 | 59.53 | **-12.35** | 8/8 | 70.7 | -3.54 |

Therefore, the overall time complexity of *MixQuant* is, which is linear with respect to the number of layers:

$$\mathcal{O}(L * B) + \mathcal{O}(L(B * M)) = \mathcal{O}(L(B + B * M)) \qquad (2)$$

If we used model loss instead of layer QE to search for optimal bits, we would need to consider all the models generated via the combinations of B number of bit-widths over L number of layers. The time complexity would be $\mathcal{O}(B^L)$, which is exponential.

## 5 RESULTS

We implement *MixQuant* using Python and combine it with two types of quantization techniques: (1) BRECQ (Li et al., 2021), a state-of-the-art quantization method, and (2) vanilla asymmetric quantization (Jacob et al., 2018) and evaluate it on the validation set of the Imagenet ILSVRC2012 dataset. Our results demonstrate that *MixQuant* can optimize the performance of existing quantization techniques.

***MixQuant* with BRECQ** BRECQ is a state-of-the art quantization method that has been shown to outperform OMSE (Choukroun et al., 2019), AdaRound (Nagel et al., 2020), AdaQuant (Hubara et al., 2020), and Bit-Split (Wang et al., 2020), and in Table 1, we demonstrate in that when *MixQuant* is combined with BRECQ, we achieve better quantized accuracy than BRECQ alone. Additionally, in Table 2 we compare our results with (Liu et al., 2021), a state of the art mixed precision quantization technique, and show that the accuracy degradation is significantly greater in Liu et al. (2021).

***MixQuant* with Asymmetric Quantization** In addition to BRECQ, we combine *MixQuant* with asymmetric quantization and compare its quantized model accuracy with f32 and int8 baselines. Table 3 shows the set of bit-widths found via *MixQuant* for various QEMs and various ResNet architectures along with model top-1 and top-5 accuracy. A user can flexibly select the quantization solution based on *their* requirements with the QEM. For higher QEMs the bit-widths are lower and the model accuracy decreases, while for lower QEMs the bit-widths and quantized model accuracy are higher. Therefore, *MixQuant* allows its user to flexibly select the trade-off between model

Table 3: *MixQuant* Results: quantization bit-widths, quantized model accuracy, loss and quantization mean squared error for various quantitative error multipliers

| Architecture | Experiment | QEM | layers_bit_widths | Acc@1 | Acc@5 | loss_avg | QMSE |
|---|---|---|---|---|---|---|---|
| resnet18 | baseline: f32 | N/A | all layers are float 32 | 69.76 | 89.08 | 1.25 | N/A |
| resnet18 | baseline: int8 | N/A | all layers are int 8 | 69.63 | 89.07 | 1.25 | N/A |
| resnet18 | | 2 | 6, 7 | 68.20 | 88.30 | 1.31 | 0.23 |
| resnet18 | | 3 | 5, 6, 7 | 63.96 | 85.58 | 1.51 | 0.36 |
| resnet18 | MixQuant | 3.25 | 5, 6 | 64.00 | 85.54 | 1.51 | 0.37 |
| resnet18 | | 3.3 | 4, 5, 6 | 61.29 | 83.81 | 1.64 | 0.37 |
| resnet18 | | 3.5 | 4, 6 | 53.67 | 77.78 | 2.04 | 0.38 |
| resnet34 | baseline: f32 | N/A | all layers are float 32 | 73.31 | 91.42 | 1.08 | N/A |
| resnet34 | baseline: int8 | N/A | all layers are int 8 | 73.24 | 91.39 | 1.08 | N/A |
| resnet34 | | 2 | 6, 7 | 72.35 | 90.91 | 1.12 | 0.24 |
| resnet34 | MixQuant | 3 | 4, 5, 6, 7 | 61.21 | 82.93 | 1.70 | 0.39 |
| resnet34 | | 3.25 | 4, 6 | 61.36 | 83.05 | 1.68 | 0.40 |
| resnet34 | | 3.3 | 4, 6 | 61.36 | 83.05 | 1.68 | 0.40 |
| resnet50 | baseline: f32 | N/A | all layers are float 32 | 76.13 | 92.86 | 0.96 | N/A |
| resnet50 | baseline: int8 | N/A | all layers are int 8 | 75.99 | 92.81 | 0.97 | N/A |
| resnet50 | | 2 | 6, 7 | 75.18 | 92.52 | 1.00 | 0.28 |
| resnet50 | MixQuant | 3 | 4, 5, 6 | 70.58 | 90.04 | 1.19 | 0.43 |
| resnet50 | | 3.25 | 4, 5, 6 | 50.13 | 74.29 | 2.30 | 0.45 |
| resnet101 | baseline: f32 | N/A | all layers are float 32 | 77.37 | 93.55 | 0.91 | N/A |
| resnet101 | baseline: int8 | N/A | all layers are int 8 | 77.21 | 93.51 | 0.92 | N/A |
| resnet101 | | 1.3 | 5, 6, 7, 8 | 76.96 | 93.42 | 0.92 | 0.22 |
| resnet101 | | 1.5 | 2, 5, 6, 7, 8 | 59.23 | 81.74 | 1.83 | 0.24 |
| resnet101 | MixQuant | 1.7 | 2, 3, 4, 5, 6, 7, 8 | 58.86 | 81.05 | 1.86 | 0.30 |
| resnet101 | | 1.8 | 2, 3, 5, 6, 7 | 52.32 | 75.61 | 2.25 | 0.33 |
| resnet101 | | 1.9 | 2, 4, 5, 6, 7 | 49.36 | 72.63 | 2.44 | 0.34 |
| resnet152 | baseline: f32 | N/A | all layers are float 32 | 78.31 | 94.05 | 0.88 | N/A |
| resnet152 | baseline: int8 | N/A | all layers are int 8 | 78.31 | 94.02 | 0.88 | N/A |
| resnet152 | | 1.1 | 7, 8 | 78.20 | 94.01 | 0.88 | 0.20 |
| resnet152 | | 1.3 | 6, 7, 8 | 78.15 | 94.01 | 0.89 | 0.20 |
| resnet152 | MixQuant | 1.5 | 5, 6, 7, 8 | 77.58 | 93.76 | 0.91 | 0.23 |
| resnet152 | | 1.7 | 3, 5, 6, 7, 8 | 70.68 | 90.11 | 1.22 | 0.28 |
| resnet152 | | 1.8 | 2, 5, 6, 7 | 71.48 | 90.16 | 1.19 | 0.31 |
| resnet152 | | 1.9 | 2, 4, 5, 6, 7 | 62.99 | 85.01 | 1.66 | 0.32 |
| resnext50_32x4d | baseline: f32 | N/A | all layers are float 32 | 77.62 | 93.70 | 0.94 | N/A |
| resnext50_32x4d | baseline: int8 | N/A | all layers are int 8 | 77.40 | 93.63 | 0.95 | N/A |
| resnext50_32x4d | | 1.3 | 7, 8 | 77.43 | 93.52 | 0.95 | 0.19 |
| resnext50_32x4d | | 1.5 | 6, 7, 8 | 77.21 | 93.51 | 0.95 | 0.20 |
| resnext50_32x4d | MixQuant | 1.7 | 5, 6, 7, 8 | 76.93 | 93.29 | 0.98 | 0.27 |
| resnext50_32x4d | | 1.8 | 5, 6, 7 | 75.43 | 92.60 | 1.05 | 0.30 |
| resnext50_32x4d | | 1.9 | 5, 6, 7 | 75.43 | 92.60 | 1.05 | 0.30 |
| resnext50_32x4d | | 2 | 4, 5, 6, 7 | 72.60 | 90.79 | 1.18 | 0.30 |
| resnext101_32x8d | baseline: f32 | N/A | all layers are float 32 | 79.31 | 94.53 | 0.93 | N/A |
| resnext101_32x8d | baseline: int8 | N/A | all layers are int 8 | 79.11 | 94.51 | 0.93 | N/A |
| resnext101_32x8d | | 1.1 | 7, 8 | 79.12 | 94.51 | 0.93 | 0.31 |
| resnext101_32x8d | | 1.3 | 4, 6, 7, 8 | 76.61 | 93.26 | 1.04 | 0.33 |
| resnext101_32x8d | MixQuant | 1.5 | 2, 4, 5, 6, 7, 8 | 59.91 | 81.46 | 2.05 | 0.39 |
| resnext101_32x8d | | 1.7 | 2, 4, 5, 6, 7, 8 | 37.65 | 59.52 | 3.84 | 0.46 |
| resnext101_32x8d | | 1.8 | 2, 3, 4, 5, 6, 7 | 26.14 | 45.57 | 5.02 | 0.49 |

accuracy and lowering the quantization bit-width. For example, the highlighted lines in Table 3 satisfy the requirement of selecting the minimum quantization bit-widths such that the model top-1 accuracy degradation is ≤ 3%.

**Runtime Analysis** Table 4 reports the runtime in seconds of *MixQuant* for various ResNet architectures, where *MixQuant* considers the bit-widths of 8, 7, 6, 5, 4, 3, and 2, and one or ten different QEMs. It can be observed that the runtime grows with the number of layers since higher number of layers imply a larger search space. For one QEM, the *MixQuant* search takes between 0.1 and 0.5 seconds. If it is combined with asymmetric per-layer quantization using the optimal bit-widths

returned by the search, it takes between 1.0 and 3.2 seconds. If the number of QEMs is increased from one to ten the *MixQuant* search takes between 0.9 and 5.5 seconds, which represents a linear increase in runtime.

## 6 QUANTIZATION SENSITIVITY OF WEIGHTS ABLATION STUDIES

To demonstrate that quantizing DNN weights warrants a search for optimal bit-widths as opposed to uniform precision quantization, we perform two ablation studies to show that different weight layers have different sensitivity to quantization based on their type and position.

**Weights Quantization Sensitivity by Layer Type**    First, we investigate if different layer types have different sensitivity to quantization. We consider four layer types in the ResNet architecture: (1) first conv layer, (2) conv layers with a 3x3 kernel, (3) conv layers with a 1x1 kernel, and (4) final fully connected layer. For each type of layer, we perform asymmetric quantization and vary its bit-width while keeping the bit-width of all other layer types constant at int8. We calculate the model accuracy, loss, and quantization error for the following quantization bit-widths: 8, 7, 6, 5, 4, 3, and 2.

In Figure 1, we show the impact of varying the bit-width of one layer type at a time on the model top-1 accuracy. Lowering the quantization bit-width of conv layers with a 3x3 kernel has the most adverse impact on top-1 accuracy in shallower ResNet architectures, while in deeper ones it is the conv layers with a 1x1 kernel followed by conv layers with a 3x3 kernel that impacts model accuracy the most. The first conv layer and conv layers with a 1x1 kernel have approximately the same sensitivity to varying bit-width in the shallower architectures. Finally, the quantization bit-width of the final fully connected layer has the smallest impact on model accuracy for all ResNet architectures. In general, starting at 5 bits the model accuracy begins to degrade; however, the deeper architectures are less sensitive to decreasing bit-width. While the reason that the conv layers with a 3x3 kernel and 1x1 kernel are the most sensitive is the fact that those layer types account for the highest number of layers in ResNet, we can still conclude that different layer types have different sensitivity to quantization bit-width measured as the impact on the overall model quality. Therefore, different layer types will benefit from different quantization bit-widths, which motivates MixQuant. Similar results can also be found by measuring layer type sensitivity using the model average loss and quantization mean squared error (Figures 2 and 3 in Appendix).

**Weights Quantization Sensitivity by Layer Position**    In addition to the layer type, we investigate if the position of a layer has an impact on quantization sensitivity of weights. We measure the *relative quantization error* (RQE) of individual layers for the following bit-widths: 8, 7, 6, 5, 4, 3, 2, and define the RQE as $RQE = avg((f32\,\vec{w} - dequan\vec{tized}\,w)/f32\,\vec{w})$, where $\vec{w}$ is the weights vector and the $avg$ operation returns a scalar that represents the mean of all elements in a vector.

Table 5 identifies the most sensitive layers across various bit-widths and architectures, where layers are indexed from 0 through n, and n equals is the total number of layers in an architecture minus one. For example, for int8, it is the 1st layer in resnet18 that has the highest relative quantization error compared to all other resnet18 layers while for resnet50 it is the 46th layer. We can see that the quantization bit-width has a significant impact on the position of the most sensitive layer with the exception of ResNet50. While ResNet50's most sensitive layer is located towards the end of

Table 4: Runtime of *MixQuant* search and *MixQuant* combined with asymmetric quantization reported in seconds for (a) 1 QEM and (b) 10 QEMs

|  | (a) |  |  | (b) |  |
|---|---|---|---|---|---|
| **Architecture** | **search** | **search + quantization** | **Architecture** | **search** | **search+quantization** |
| resnet18 | 0.1 s | 1 s | resnet18 | 0.9 s | 1.8 s |
| resnet34 | 0.2 s | 1.1 s | resnet34 | 1.5 s | 2.5 s |
| resnet50 | 0.2 s | 1.3 s | resnet50 | 2 s | 3.1 s |
| resnet101 | 0.4 s | 1.7 s | resnet101 | 3.6 s | 4.9 s |
| resnet152 | 0.5 s | 2 s | resnet152 | 5.2 s | 6.8 s |
| resnext50_32x4d | 0.2 s | 1.4 s | resnext50_32x4d | 2 s | 3.2 s |
| resnext101_32x8d | 0.5 s | 3.2 s | resnext101_32x8d | 5.5 s | 8.2 s |

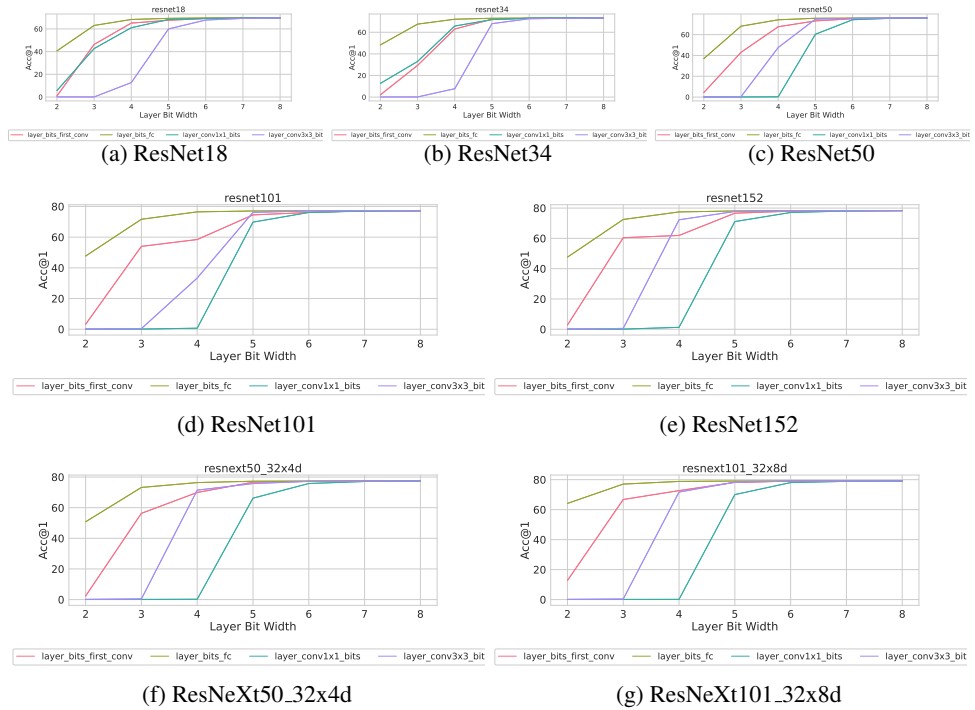

Figure 1: Sensitivity of different layer types to quantization measured as quantized model top-1 accuracy with respect to varying bit-width of one type of layer while holding all other layer types bit-widths constant at int8

Table 5: The most sensitive layer positions in a DNN measured as a relative quantization error with respect to varying quantization bit-width

| Architecture | Most sensitive layer position at various quantization bit-widths | | | | | | |
|---|---|---|---|---|---|---|---|
| | int 8 | int 7 | int 6 | int 5 | int 4 | int 3 | int 2 |
| resenet18 | 1 | 1 | 1 | 17 | 17 | 17 | 16 |
| resnet34 | 1 | 1 | 20 | 20 | 20 | 33 | 35 |
| resnet50 | 46 | 46 | 46 | 46 | 46 | 47 | 44 |
| resnet101 | 6 | 6 | 6 | 6 | 97 | 97 | 99 |
| resnet152 | 1 | 45 | 45 | 148 | 148 | 148 | 152 |
| resnext50_32x4d | 1 | 1 | 1 | 52 | 45 | 45 | 45 |
| resnext101_32x8d | 1 | 1 | 49 | 49 | 49 | 96 | 96 |

the network for all quantization bit-widths, other architectures's most sensitive layer position varies based on the bit-width. For higher bit-widths 8, 7, and 6 it is located at the beginning while for lower bit-widths 2, 3, and 4 it is at the end. The most sensitive layers of ResNet34 and ResNeXt101_32x8d at bit-widths 4, 5, and 6 are positioned in the middle of the network. Based on these experiments, we can conclude that different layer positions have different sensitivity to varying bit-width. Additionally, we can see that the position of sensitive layers depends on the bit-width and network architecture.

## 7 CONCLUSION

In this paper we propose *MixQuant*, a search algorithm that finds the optimal quantization bit-width for each layer weight and can be combined with any quantization method as a form of pre-processing optimization. We show that combining *MixQuant* with BRECQ (Li et al., 2021), a state-of-the-art quantization method, yields better quantized model accuracy than BRECQ alone. Additionally, we combine BREQ with asymmetric quantization (Jacob et al., 2018) to show that *MixQuant* has the potential to optimize the performance of any quantization technique. Our code is open-sourced and available at: https://anonymous.4open.science/r/qantizedImagenet-43C5.

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

# A APPENDIX

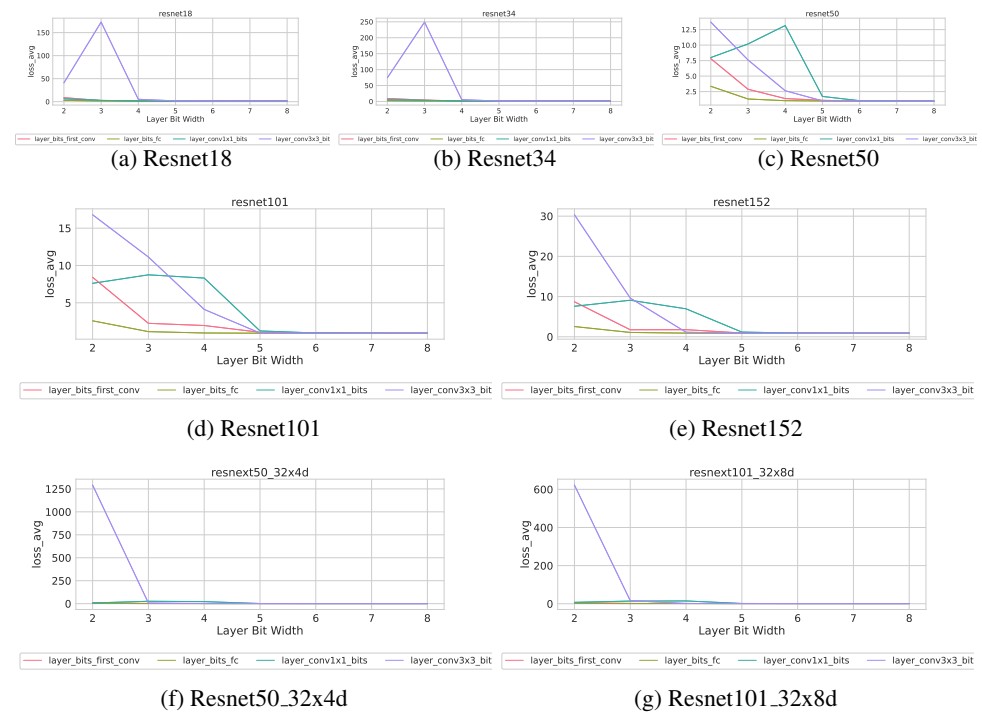

Figure 2: Sensitivity of different layer types to quantization by architecture measured as quantized model average loss with respect to varying bit-width of one type of layer while holding all other layer types bit-widths constant at int 8

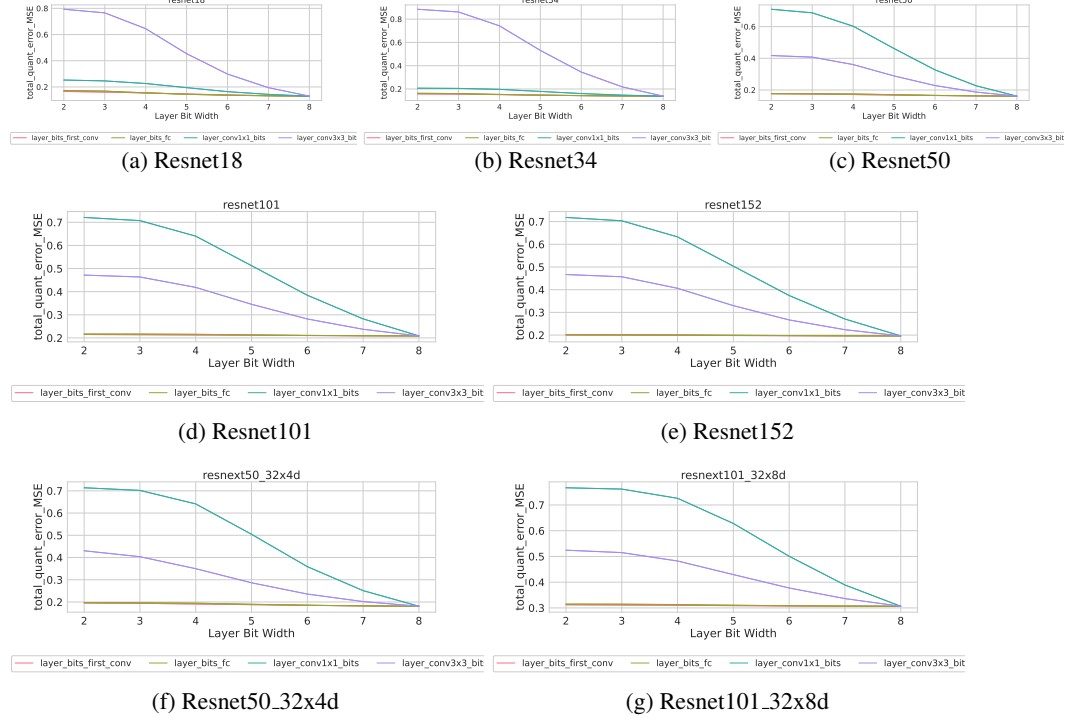

Figure 3: Sensitivity of different layer types to quantization by architecture measured as quantized model total quantization mean squared error with respect to varying bit-width of one type of layer while holding all other layer types bit-widths constant at int 8

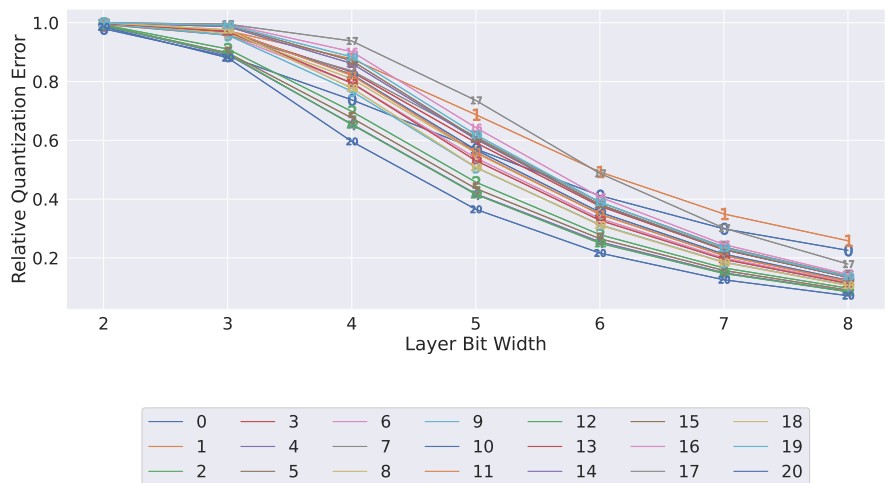

Figure 4: Resnet18 layer sensitivity with respect to quantization bit-width

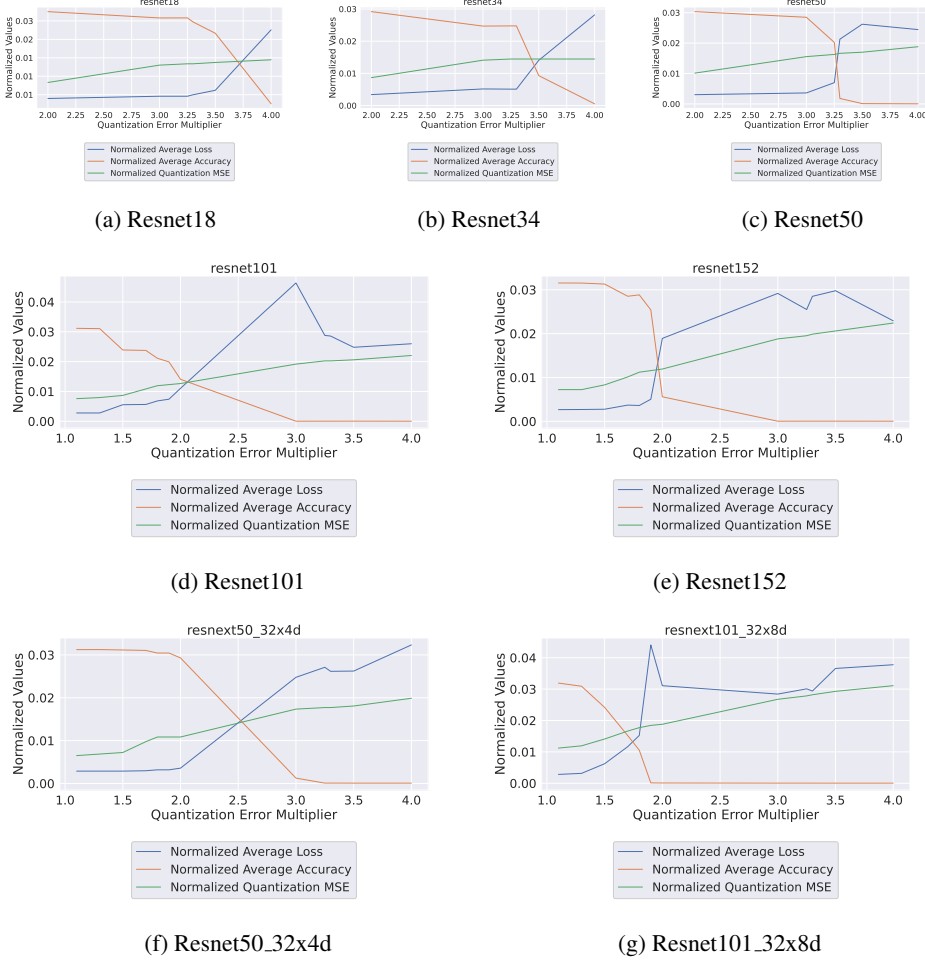

Figure 5: Relationship between quantization error multiplier (QEM) and model accuracy by architecture

Table 6: *MixQuant* results for ResNet18, ResNet34 and ResNet50: individual layer quantization bit-width assignments, quantized model accuracy, loss and quantization mean squared error for various quantitative error multipliers

| arch | QEM | layers_bit_widths | Acc@1 | Acc@5 | loss_avg | QMSE |
|---|---|---|---|---|---|---|
| resnet18 | N/A | all layers are float 32 | 69.76 | 89.08 | 1.25 | N/A |
| resnet18 | N/A | all layers are int 8 | 69.63 | 89.07 | 1.25 | N/A |
| resnet18 | 2 | [6, 6, 7, 7, 7, 7, 7, 7, 7, 7, 7, 7, 7, 7, 7, 7, 7, 7, 7, 7] | 68.20 | 88.30 | 1.31 | 0.23 |
| resnet18 | 3 | [5, 5, 6, 6, 6, 6, 6, 6, 6, 6, 6, 6, 6, 6, 6, 6, 6, 6, 6, 7] | 63.96 | 85.58 | 1.51 | 0.36 |
| resnet18 | 3.25 | [5, 5, 6, 6, 6, 6, 6, 6, 6, 6, 6, 6, 6, 6, 6, 6, 6, 6, 6, 6] | 64.00 | 85.54 | 1.51 | 0.37 |
| resnet18 | 3.3 | [4, 5, 6, 6, 6, 6, 6, 6, 6, 6, 6, 6, 6, 6, 6, 6, 6, 6, 6, 6] | 61.29 | 83.81 | 1.64 | 0.37 |
| resnet18 | 3.5 | [4, 4, 6, 6, 6, 6, 6, 6, 6, 6, 6, 6, 6, 6, 6, 6, 6, 6, 6, 6] | 53.67 | 77.78 | 2.04 | 0.38 |
| resnet34 | N/A | all layers are float 32 | 73.31 | 91.42 | 1.08 | N/A |
| resnet34 | N/A | all layers are int 8 | 73.24 | 91.39 | 1.08 | N/A |
| resnet34 | 2 | [6, 6, 7, 7, 7, 7, 7, 7, 7, 7, 7, 7, 7, 7, 7, 7, 7, 7, 7, 7, 7, 7, 7, 7, 7, 7, 7, 7, 7, 7, 7, 7, 7, 7, 7, 7] | 72.35 | 90.91 | 1.12 | 0.24 |
| resnet34 | 3 | [4, 5, 6, 6, 6, 6, 6, 6, 6, 6, 6, 6, 6, 6, 6, 6, 6, 6, 6, 6, 6, 6, 6, 6, 6, 6, 6, 6, 6, 6, 6, 6, 6, 6, 6, 7] | 61.21 | 82.93 | 1.70 | 0.39 |
| resnet34 | 3.25 | [4, 4, 6, 6, 6, 6, 6, 6, 6, 6, 6, 6, 6, 6, 6, 6, 6, 6, 6, 6, 6, 6, 6, 6, 6, 6, 6, 6, 6, 6, 6, 6, 6, 6, 6, 6] | 61.36 | 83.05 | 1.68 | 0.40 |
| resnet34 | 3.3 | [4, 4, 6, 6, 6, 6, 6, 6, 6, 6, 6, 6, 6, 6, 6, 6, 6, 6, 6, 6, 6, 6, 6, 6, 6, 6, 6, 6, 6, 6, 6, 6, 6, 6, 6, 6] | 61.36 | 83.05 | 1.68 | 0.40 |
| resnet50 | N/A | all layers are float 32 | 76.13 | 92.86 | 0.96 | N/A |
| resnet50 | N/A | all layers are int 8 | 75.99 | 92.81 | 0.97 | N/A |
| resnet50 | 2 | [7, 6, 6, 6, 6, 6, 7, 7, 7, 7, 7, 7, 7, 6, 6, 6, 7, 7, 7, 7, 7, 7, 7, 7, 7, 7, 7, 7, 7, 7, 7, 7, 7, 7, 7, 7, 7, 7, 7, 7, 7, 7, 7, 7, 7, 7, 7, 7, 7, 7, 7, 7, 7] | 75.18 | 92.52 | 1.00 | 0.28 |
| resnet50 | 3 | [6, 5, 5, 4, 5, 5, 6, 6, 6, 6, 6, 6, 6, 5, 5, 5, 6, 6, 6, 6, 6, 6, 6, 6, 6, 6, 6, 6, 6, 6, 6, 6, 6, 6, 6, 6, 6, 6, 6, 6, 6, 6, 6, 6, 6, 6, 6, 6, 6, 6, 6, 6, 6] | 70.58 | 90.04 | 1.19 | 0.43 |
| resnet50 | 3.25 | [6, 5, 4, 4, 4, 4, 6, 5, 6, 6, 6, 6, 6, 5, 4, 5, 6, 5, 6, 6, 6, 6, 6, 6, 6, 6, 6, 6, 6, 6, 6, 6, 6, 6, 6, 6, 6, 6, 6, 6, 6, 6, 6, 2, 6, 6, 6, 6, 6, 6, 6, 6] | 50.13 | 74.29 | 2.30 | 0.45 |

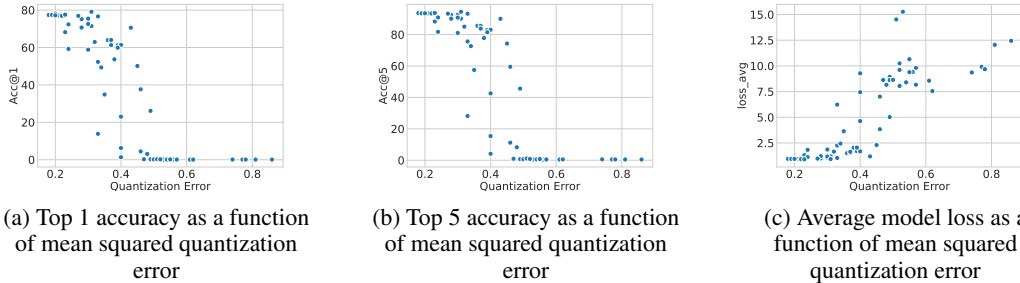

(a) Top 1 accuracy as a function of mean squared quantization error

(b) Top 5 accuracy as a function of mean squared quantization error

(c) Average model loss as a function of mean squared quantization error

Figure 6: Relationship between quantization error and model accuracy & loss

Table 7: *MixQuant* results for ResNet101 and ResNet152: individual layer quantization bit-width assignments, quantized model accuracy, loss and quantization mean squared error for various quantitative error multipliers

| arch | QEM | layers_bit_widths | Acc@1 | Acc@5 | loss_avg | QMSE |
|------|-----|-------------------|-------|-------|----------|------|
| resnet101 | N/A | all layers are float 32 | 77.37 | 93.55 | 0.91 | N/A |
| resnet101 | N/A | all layers are int 8 | 77.21 | 93.51 | 0.92 | N/A |
| resnet101 | 1.3 | [7, 6, 7, 8, 7, 6, 5, 7, 8, 8, 8, 8, 8, 8, 8, 8, 7, 8, 8, 8, 8, 8, 8, 8, 8, 8, 8, 8, 8, 8, 8, 8, 8, 8, 8, 8, 8, 8, 8, 8, 8, 8, 8, 8, 8, 8, 8, 8, 8, 8, 8, 8, 8, 8, 8, 8, 8, 8, 8, 8, 8, 8, 8, 8, 8, 8, 8, 8, 8, 8, 8, 8, 8, 8, 8, 8, 8, 8, 8, 8, 8, 8, 8, 8, 8, 8, 8, 8, 8, 8, 8, 8, 8, 8, 8, 8, 8, 8, 8, 8, 8] | 76.96 | 93.42 | 0.92 | 0.22 |
| resnet101 | 1.5 | [7, 6, 7, 7, 6, 5, 2, 6, 7, 7, 8, 8, 8, 7, 7, 7, 8, 6, 7, 8, 8, 8, 8, 8, 8, 8, 8, 8, 8, 8, 7, 7, 7, 8, 7, 7, 8, 7, 8, 8, 7, 8, 8, 7, 8, 8, 7, 8, 8, 8, 8, 8, 8, 8, 8, 8, 8, 8, 8, 8, 8, 8, 8, 8, 8, 8, 8, 8, 8, 8, 8, 8, 8, 8, 8, 8, 8, 8, 8, 8, 8, 8, 8, 8, 8, 8, 8, 8, 8, 8, 8, 8, 8, 8, 8, 8, 8, 8, 8, 8, 8] | 59.23 | 81.74 | 1.83 | 0.24 |
| resnet101 | 1.7 | [6, 5, 6, 7, 6, 4, 2, 3, 7, 7, 8, 8, 7, 7, 7, 7, 5, 7, 7, 7, 8, 7, 8, 8, 7, 7, 7, 7, 6, 7, 7, 7, 7, 8, 7, 7, 7, 7, 7, 8, 7, 7, 7, 8, 7, 7, 8, 7, 7, 8, 7, 8, 8, 7, 8, 8, 7, 8, 8, 7, 8, 8, 7, 8, 7, 8, 8, 7, 7, 8, 8, 7, 7, 8, 8, 7, 7, 8, 7, 8, 8] | 58.86 | 81.05 | 1.86 | 0.30 |
| resnet101 | 1.8 | [6, 5, 6, 7, 5, 3, 2, 2, 6, 7, 7, 7, 6, 7, 7, 7, 5, 7, 7, 7, 7, 7, 7, 7, 7, 7, 7, 6, 7, 7, 7, 7, 7, 7, 7, 7, 6, 7, 7, 7, 7, 7, 7, 7, 7, 7, 7, 7, 7, 7, 7, 7, 7, 7, 7, 7, 7, 7, 7, 7, 7, 7, 7, 7, 7, 7, 7, 7, 7, 7, 7, 7, 7, 7, 7, 7, 7, 7, 7, 7, 7, 7, 7, 7, 7, 7, 7, 7, 7, 7, 7, 7, 7, 7, 7] | 52.32 | 75.61 | 2.25 | 0.33 |
| resnet101 | 1.9 | [6, 4, 6, 6, 5, 2, 2, 2, 6, 7, 7, 7, 6, 6, 7, 4, 7, 7, 7, 7, 7, 7, 7, 7, 6, 7, 6, 7, 7, 6, 7, 7, 6, 7, 7, 6, 7, 7, 7, 7, 7, 7, 7, 7, 7, 7, 7, 7, 7, 7, 7, 7, 7, 7, 7, 7, 7, 7, 7, 7, 7, 7, 7, 7, 7, 7, 7, 7, 7, 7, 7, 7, 7, 7, 7, 7, 7, 7, 7, 7, 7] | 49.36 | 72.63 | 2.44 | 0.34 |
| resnet152 | N/A | all layers are float 32 | 78.31 | 94.05 | 0.88 | N/A |
| resnet152 | N/A | all layers are int 8 | 78.31 | 94.02 | 0.88 | N/A |
| resnet152 | 1.3 | [7, 6, 7, 7, 7, 7, 8, 7, 7, 8, 8, 8, 8, 8, 8, 8, 7, 7, 8, 8, 8, 8, 8, 8, 7, 7, 8, 8, 8, 8, 8, 8, 8, 8, 8, 8, 8, 8, 8, 8, 8, 8, 8, 8, 8, 8, 8, 8, 8, 8, 8, 8, 8, 8, 8, 8, 8, 8, 8, 8, 8, 8, 8, 8, 8, 8, 8, 8, 8, 8, 8, 8, 8, 8, 8, 8, 8, 8, 8, 8, 8, 8, 8, 8, 8, 8, 8, 8, 8, 8, 8, 8, 8, 8, 8, 8, 8, 8, 8, 8, 8, 8, 8, 8, 8, 8, 8] | 78.15 | 94.01 | 0.89 | 0.20 |
| resnet152 | 1.5 | [6, 5, 6, 7, 6, 6, 7, 7, 7, 7, 8, 7, 7, 7, 7, 7, 7, 7, 7, 7, 7, 7, 8, 7, 7, 7, 7, 6, 7, 7, 8, 8, 8, 7, 8, 8, 7, 8, 8, 7, 7, 8, 7, 7, 8, 7, 8, 8, 8, 8, 7, 8, 8, 8, 8, 8, 7, 8, 8, 8, 8, 8, 7, 8, 8, 8, 8, 8, 8, 8, 8, 8, 8, 8, 8, 8, 8, 8, 8, 8, 8, 8, 8, 8, 8, 8, 8, 8, 8, 8, 8, 8, 8, 8, 8, 8, 8, 8, 8, 8, 8, 8, 8, 8, 8, 8, 8] | 77.58 | 93.76 | 0.91 | 0.23 |
| resnet152 | 1.7 | [6, 3, 5, 6, 5, 6, 6, 7, 6, 7, 7, 7, 7, 7, 7, 7, 7, 7, 7, 6, 7, 7, 7, 7, 7, 7, 7, 7, 6, 7, 7, 7, 8, 8, 8, 7, 7, 7, 7, 7, 7, 7, 7, 7, 8, 7, 8, 7, 7, 7, 8, 7, 7, 7, 8, 7, 7, 8, 7, 7, 8, 7, 7, 8, 7, 7, 8, 7, 7, 8, 7, 7, 8, 8, 7, 7, 8, 7, 8, 7, 8, 8, 7, 8, 7, 7, 8, 8, 7, 8, 8, 7, 8, 7, 8, 8, 7, 8, 8, 8, 7, 8, 8, 8, 7, 8, 8, 7, 8, 8, 8, 7, 8, 8, 8] | 70.68 | 90.11 | 1.22 | 0.28 |
| resnet152 | 1.8 | [5, 2, 5, 6, 5, 5, 6, 7, 6, 6, 7, 7, 7, 7, 7, 7, 6, 6, 7, 7, 7, 7, 7, 7, 6, 6, 7, 7, 7, 7, 7, 7, 7, 7, 6, 6, 7, 7, 7, 7, 7, 7, 7, 7, 7, 7, 7, 7, 7, 7, 7, 7, 7, 7, 7, 7, 7, 7, 7, 7, 7, 7, 7, 7, 7, 7, 7, 7, 7, 7, 7, 7, 7, 7, 7, 7, 7, 7, 7, 7, 7, 7, 7, 7, 7, 7, 7, 7, 7, 7, 7, 7, 7, 7, 7, 7, 7, 7, 7, 7, 7, 7, 7, 7, 7, 7] | 71.48 | 90.16 | 1.19 | 0.31 |
| resnet152 | 1.9 | [5, 2, 4, 6, 4, 5, 6, 6, 6, 6, 7, 7, 7, 7, 7, 7, 6, 6, 6, 7, 7, 6, 7, 7, 6, 7, 7, 6, 7, 6, 7, 6, 7, 6, 6, 5, 7, 7, 7, 7, 7, 6, 7, 7, 7, 7, 7, 6, 7, 6, 7, 7, 6, 7, 7, 7, 7, 7, 7, 7, 7, 7, 7, 7, 7, 7, 7, 7, 7, 7, 7, 7, 7, 7, 7, 7, 7, 7, 7, 7, 7, 7, 7, 7, 7, 7, 7, 7, 7, 7, 7, 7, 7, 7, 7, 7, 7, 7, 7, 7, 7, 7, 7, 7, 7, 7, 7] | 62.99 | 85.01 | 1.66 | 0.32 |

Table 8: *MixQuant* results for ResNeXt50_32x4d and ResNeXt101_32x8d: individual layer quantization bit-width assignments, quantized model accuracy, loss and quantization mean squared error for various quantitative error multipliers

| arch | QEM | layers_bit_widths | Acc@1 | Acc@5 | loss_avg | QMSE |
|------|-----|-------------------|-------|-------|----------|------|
| resnext50 32x4d | N/A | all layers are float 32 | 77.62 | 93.70 | 0.94 | N/A |
| resnext50 32x4d | N/A | all layers are int 8 | 77.40 | 93.63 | 0.95 | N/A |
| resnext50 32x4d | 1.3 | [8, 7, 7, 7, 7, 7, 7, 8, 7, 8, 8, 8, 8, 8, 8, 8, 8, 8, 8, 8, 8, 8, 8, 8, 8, 8, 8, 8, 8, 8, 8, 8, 8, 8, 8, 8, 8, 8, 8, 8, 8, 8, 8, 8, 8, 8, 8, 8, 8] | 77.43 | 93.52 | 0.95 | 0.19 |
| resnext50 32x4d | 1.5 | [8, 6, 6, 7, 7, 6, 6, 7, 7, 7, 7, 8, 8, 8, 8, 8, 8, 8, 8, 8, 8, 8, 8, 8, 8, 8, 8, 8, 8, 8, 8, 8, 8, 8, 8, 8, 8, 8, 8, 8, 8, 8, 8, 8, 8, 8, 8, 8, 8] | 77.21 | 93.51 | 0.95 | 0.20 |
| resnext50 32x4d | 1.7 | [7, 6, 6, 6, 6, 5, 6, 6, 7, 7, 7, 7, 7, 7, 7, 7, 7, 7, 8, 7, 8, 7, 7, 8, 8, 7, 7, 7, 7, 7, 8, 7, 8, 8, 7, 8, 8, 7, 7, 7, 7, 7, 7, 8, 8, 8, 7, 7, 7, 8] | 76.93 | 93.29 | 0.98 | 0.27 |
| resnext50 32x4d | 1.8 | [7, 5, 5, 6, 6, 5, 5, 6, 7, 6, 7, 7, 7, 7, 7, 7, 7, 7, 7, 7, 7, 7, 7, 7, 7, 7, 7, 7, 7, 7, 7, 7, 7, 7, 7, 7, 7, 7, 7, 7, 7, 7, 7, 7, 7, 7, 7, 7, 7] | 75.43 | 92.60 | 1.05 | 0.30 |
| resnext50 32x4d | 1.9 | [7, 5, 5, 6, 6, 5, 5, 6, 7, 6, 7, 7, 7, 7, 7, 7, 7, 7, 7, 7, 7, 7, 7, 7, 7, 7, 7, 7, 7, 7, 7, 7, 7, 7, 7, 7, 7, 7, 7, 7, 7, 7, 7, 7, 7, 7, 7, 7, 7] | 75.43 | 92.60 | 1.05 | 0.30 |
| resnext50 32x4d | 2 | [7, 4, 5, 6, 6, 4, 5, 6, 7, 6, 7, 7, 7, 7, 7, 7, 7, 7, 7, 7, 7, 7, 7, 7, 7, 7, 7, 7, 7, 7, 7, 7, 7, 7, 7, 7, 7, 7, 7, 7, 7, 7, 7, 7, 7, 7, 7, 7, 7] | 72.60 | 90.79 | 1.18 | 0.30 |
| resnext10 32x8d | N/A | all layers are float 32 | 79.31 | 94.53 | 0.93 | N/A |
| resnext101 32x8d | N/A | all layers are int 8 | 79.11 | 94.51 | 0.93 | N/A |
| resnext101 32x8d | 1.1 | [8, 7, 7, 8, 8, 8, 7, 8, 8, 7, 8, 8, 8, 8, 8, 8, 8, 8, 8, 8, 8, 8, 8, 8, 8, 8, 8, 8, 8, 8, 8, 8, 8, 8, 8, 8, 8, 8, 8, 8, 8, 8, 8, 8, 8, 8, 8, 8, 8, 8, 8, 8, 8, 8, 8, 8, 8, 8, 8, 8, 8, 8, 8, 8, 8, 8, 8, 8, 8, 8, 8, 8, 8, 8, 8, 8, 8, 8, 8, 8, 8, 8, 8] | 79.12 | 94.51 | 0.93 | 0.31 |
| resnext101 32x8d | 1.3 | [7, 4, 6, 7, 7, 6, 6, 7, 6, 6, 7, 8, 8, 8, 8, 7, 7, 8, 7, 8, 8, 7, 8, 8, 8, 8, 8, 8, 8, 8, 8, 8, 8, 8, 8, 8, 8, 8, 8, 8, 8, 8, 8, 8, 8, 8, 8, 8, 8, 8, 8, 8, 8, 8, 8, 8, 8, 8, 8, 8, 8, 8, 8, 8, 8, 8, 8, 8, 8, 8, 8, 8, 8, 8, 8, 8, 8, 8, 8] | 76.61 | 93.26 | 1.04 | 0.33 |
| resnext101 32x8d | 1.5 | [6, 2, 4, 7, 6, 5, 5, 6, 5, 5, 7, 7, 7, 8, 8, 7, 6, 7, 7, 7, 7, 7, 7, 7, 8, 8, 8, 8, 7, 7, 7, 7, 7, 7, 7, 7, 7, 7, 7, 7, 7, 7, 7, 7, 7, 7, 7, 7, 7, 7, 7, 8, 7, 8, 8, 7, 8, 7, 8, 7, 8, 8, 7, 8, 8, 8, 7, 8, 8, 8, 8, 8, 8, 8, 8, 8, 8, 8, 8, 8, 8, 8, 8] | 59.91 | 81.46 | 2.05 | 0.39 |
| resnext101 32x8d | 1.7 | [5, 2, 2, 6, 5, 4, 4, 6, 5, 4, 6, 7, 7, 7, 7, 7, 6, 6, 7, 6, 7, 7, 6, 7, 7, 7, 7, 7, 7, 7, 6, 6, 7, 6, 7, 7, 6, 7, 7, 7, 7, 6, 7, 7, 6, 6, 6, 6, 7, 7, 6, 7, 7, 7, 7, 7, 7, 7, 7, 7, 7, 7, 7, 7, 7, 7, 7, 7, 7, 7, 7, 7, 7, 7, 7, 7, 7, 8, 7, 8, 8, 7, 8] | 37.65 | 59.52 | 3.84 | 0.46 |
| resnext101 32x8d | 1.8 | [5, 2, 2, 6, 5, 2, 3, 5, 4, 3, 6, 7, 7, 7, 7, 7, 6, 6, 7, 6, 7, 7, 6, 7, 7, 7, 7, 7, 6, 6, 7, 6, 6, 6, 7, 6, 6, 6, 6, 7, 6, 7, 7, 6, 6, 6, 6, 6, 6, 6, 6, 7, 6, 7, 7, 6, 7, 7, 6, 7, 7, 6, 7, 7, 7, 7, 7, 7, 7, 7, 7, 7, 7, 7, 7, 7, 7, 7, 7, 7, 7, 7, 7] | 26.14 | 45.57 | 5.02 | 0.49 |

