# OpenReview forum: "MixQuant: A Quantization Bit-width Search that Can Optimize the Performance of your Quantization Method"
_ICLR.cc/2023/Conference — Submitted to ICLR 2023_

### Official Review · Reviewer_MUWW · 2022-10-24

**Confidence:** 4
**Correctness:** 3
**Technical Novelty And Significance:** 2
**Empirical Novelty And Significance:** 2
**Recommendation:** 3

**Clarity, Quality, Novelty And Reproducibility:**

Clarity
The problem setup and motivation are clear. The paper also provides details about its methodology.

Quality
The writing quality of this paper overall is decent and is easy-to-follow.

Novelty
The paper appears to be of limited novelty. Most of the proposed techniques seem to have been proposed by prior works. The paper did not do a thorough literature search, missing references to related work and describing observations that seem to be common understanding for model quantization.

Reproducibility
The paper provides a limited description of its implementation and hyperparameter settings, making reproducing its results possible but not very easy.


**Strength And Weaknesses:**

Strengths:
- Thorough evaluation of mixed-precision quantization on variants of ResNets.
- Demonstrated that the proposed method can be composed with other methods such as BRECQ while leading to improved performance.

Weaknesses:
- Most findings from this paper seem to be well-known facts that have been explored by prior works. So the novelty of the proposed method is limited. For example, why is it a new observation that "quantizing and dequantizing the model parameters lead to roundoff errors" and "some weights are more sensitive than others which should be reflected on their quantization bi-width"? The former phenomenon is pretty much observed by all quantization-based studies, and the latter one is what motivates all prior studies on mixed-precision quantization.  The main idea is to quantize the model layer-by-layer by solving a minimization problem of quantization error for each individual layer. Such a scheme has been explored by prior works such as [1] and [2]. The idea that different weights may be replaced with values of varying bit-width is also not new, as it has been studied by [3] and [4]. Several of these prior works are not references, which also indicates that the authors did not do a thorough literature search.
- The proposed method only considers mixed precision for weights whereas the activations remain to be 32. As such, there is no actual performance improvement from the proposed method because the actual computation still happens in FP32.
- The evaluation scope is quite limited, focusing primarily on image classification tasks and particularly ResNet variations. The authors are encouraged to evaluate its approach on a wider range of tasks/datasets, such as including NLP models and Transformer-based models such as vision transformers.
- Quantizing bits to INT2, INT3, INT5, INT6, and INT7 cannot lead to any performance gains, to the best of the reviewer's knowledge, as there is no real hardware that can benefit from this irregular bit-width.

[1] Hassibi et. al. "Optimal Brain Surgeon and general network pruning", 1993
[2] Frantar et. al. "Optimal Brain Compression", 2022
[3] Shen et al. "Q-BERT: Hessian Based Ultra Low Precision Quantization of BER". 2020
[4] Zhao et. al. "Automatic Mixed-Precision Quantization Search of BERT", 2021


**Summary Of The Paper:**

This paper introduces a mixed-precision quantization method, called MixQuant, to identify the bit-widths of individual layer weights. In particular, the paper proposes to quantize model weights layer-by-layer by greedily minimizing the quantization error from using low bit-width values. Evaluation of the proposed method on ResNets over the Imagenet dataset shows that the proposed method can find mixed-precision quantization solutions that achieve comparable accuracy in a relatively short amount of time.

**Summary Of The Review:**

The paper has done some good evaluations and ablation studies on mixed-precision quantization using its proposed method and demonstrated how the method can be composed with existing ones. However, there are some major concerns on the technical novelty and small scope of evaluation.

---

### Official Review · Reviewer_6kQQ · 2022-10-24

**Confidence:** 5
**Correctness:** 2
**Technical Novelty And Significance:** 2
**Empirical Novelty And Significance:** Not applicable
**Recommendation:** 3

**Clarity, Quality, Novelty And Reproducibility:**

Given the fact that quantization has been studied extensively by the early works, this idea may have been explored by some other works before.

**Strength And Weaknesses:**

+ Algorithm 1 is basically the core of the paper. The method is technically sound and clear
+ The writing is clear in general
- The method is very simple and lack of originality. Algorithm 1 simply performs exhaustive search on all the weight precisions, and find the optimal precision that can achieve comparable quantization error with the 8-bit quantization.
- The evaluation lacks some important baseline algorithms. This layerwise mix quantization has been studied extensively by the early work.
- Implementation of MixQuant DNN is difficult. How to design the hardware platform to have this mixQuant DNN working?


**Summary Of The Paper:**

This paper presents a search algorithm named MixQuant. MixQuant can search for the optimal bit-width for each layer weight based on the the l2 quantization error. Combined with BRECQ, MixQuant + BRECQ achieves the  state-of-the-art performance across multiple datasets.


**Summary Of The Review:**

Overall, I think this idea is too obvious and this problem is either not new. Originality is not enough to get this work accept.
In addition, the evaluation also lacks a lot of important baselines.
Minor problem: in forth row of section 4: "Prior works have shown ...", please add some reference for these works.

---

### Official Review · Reviewer_Kh8z · 2022-10-24

**Confidence:** 5
**Correctness:** 1
**Technical Novelty And Significance:** 1
**Empirical Novelty And Significance:** 1
**Recommendation:** 1

**Clarity, Quality, Novelty And Reproducibility:**

I think this paper should be refined for many reasons. Some statements are not clear to support their method, which seems to be not novel. It is easy to reproduce their results, but I don’t think it is meaningful for developers or researchers in this domain.

**Strength And Weaknesses:**

Strength
 - This method is simple, straightforward and easy to adopt.
 - Assigning varied bit-widths to each layer is not a time-consuming process.

Weakness
 - The quantization method is not quite novel because this method is not complex or an intelligence optimization algorithm. It seems to be just a kind of heuristic algorithm with layer-wise bit-selection with a threshold of quantization errors. There is less theoretical reason for this algorithm, and it is not proved that this algorithm can find optimal set of bit-widths to minimize quantization errors. Indeed, minimizing quantization doesn’t exactly mean the best performance of a quantized model (e.g. recent PTQ papers have pointed out MSE is not an optimal solution for quantization).
 - In the practical side of quantization, I don’t understand why this paper adopts such a quantization scheme. They quantize only weights with various bit-widths from 2-bits to 8-bits. Because there are no such processing units for that, the quantized weights should be dequantized into full-precision format during inference. So, there is no gain on computation due to their quantization. Then, they can reduce only memory-bounded problems. But, the target networks of this paper are only CNNs, which are computation-bounded models (i.e. the weights of CNNs are not a big portion of memory during inference.) Many quantization papers also include weight-only quantization results, but they also provide these results as a bridge to show their final results when quantizing both weight and activation. Since this paper seems not to be applicable to activation quantization, it is hard to extend to real computation units (INT8/4).
 - In addition, I’m very curious why this paper compares 4/8 results from (Liu, 2021) while this paper has no idea about activation quantization. Indeed, recent PTQ papers have improved the w4a8 quantization results, which are close to full-precision.


**Summary Of The Paper:**

This paper proposes an optimization algorithm for mixed-precision weight quantization. This algorithm decides the bit-width of each layer in forward order. The optimization target is the amount of quantization errors. Their method can make mixed-precision quantized networks (weight only) using various bit-widths with higher accuracy.

**Summary Of The Review:**

I think this paper and this method should be refined in the aspects of quantization format and optimization algorithm.

---

### Official Review · Reviewer_Z2i6 · 2022-10-24

**Confidence:** 4
**Correctness:** 1
**Technical Novelty And Significance:** 2
**Empirical Novelty And Significance:** 1
**Recommendation:** 1

**Clarity, Quality, Novelty And Reproducibility:**

The quality of the work is poor and has major flaws in the empirical evaluation and also the proposed algorithm and maths does not align with the text and in it’s current form would lead to a trivial solution (always 8 bits). Also the search algorithm is very simple and trivial with little novelty. Except some mistakes, the paper is clearly written but not concise (this simple idea could have easily been proposed in a 4 page workshop paper).


**Strength And Weaknesses:**


Strength:
* The search algorithm is very simple and fast (just a few seconds).
* Can be combined with other PTQ methods such as BRECQ etc.
* The paper is language wise well written and easy to follow.

Weaknesses:
* Their bit-width selection seems wrong (eq 1 and line 12 in algorithm 1): the quantization error for the highest bit-width (here 8 bits) should almost always give the lowest quantization error, thus the argmin will select this. And 8 bits will always satisfy the condition for QEM >= 1 which is the case in the experiment section. (Based on the text my guess would be that this equation might rather be $optBit = argmin_{optBit \in B} optBit$).
* The paper wrongly claims that MixQuant finds the optimal bit-widths. There is no proof for this and in general finding the optimal bit-width is NP-hard and any linear algorithm will be unable to find the optimal solution for all cases.
* Empirical evaluation is very bad and no conclusions can be drawn from them. A few examples:
    * No average bit-width or BOPs are reported for MixQuant in any table making it impossible to compare with fixed bit-widths. The alignment in the table is even misleading, e.g. in table 1 and 2 they put MixQuant with 4,5,6 bits next to fixed 4 bits and fixed 3 bits, which are clearly not comparable in size or BOPs.
    * In several cases MixQuant is better than the reported baseline (both MobileNet and Resent). This can not be true given that we do post training quantization in which only noise/error is added to the model but no training is performed, especially not almost 1% as it is for Resnet18 (70.69% vs 69.76%).
    * Several rows in table 1 and 2 are duplicated with no additional value (some rows in table 2 are literarily identical). Full table 2 could be omitted as it does not add new results, just compares to Liu et al. which could be added to table 1 (and Liu et al. seems a irrelevant baseline as it completely underperforms, even older literature such as AdaRound or BitSplit performs significantly better than that).
* Quantization and numeric instability and listing 1: The example is trivial and it does not show more than that quantization error (here rounding error) does exist in practice and also leads to and error after the matmul. That quantization error exists is logical and widely known in the field and does definitely not justify an example of over 1 page in the main paper.
* They do not do activation quantization and I do not fully agree with their reasoning. I agree somewhat with argument one, argument two is actually a reason why we should do activation quantization as well and have mixed precision for activations. The third reason is misleading as BRECQ is also doing activation quantization except in one ablation study.
* Is W a single weight or a tensor? Assuming the latter, then in line 9 of algorithm one (the quantization error) would also be a tensor. How is this error aggregated and compared? MSE?
* For the layer-wise error alternatives should be discussed and empirically considered. E.g. BRECQ/AdaRound which the authors heavily cite and compare to show that the MSE of the output activations ($||Wx - W_q x||^2_F$) is a better task loss estimate than the MSE on the weights ($||W - W_q||^2_F$) for weight quantization error. Should that not help for mixed precision as well?


**Summary Of The Paper:**

The authors propose MixQuant, a simple mixed precision search algorithm. MixQaunt evaluates the quantization error per layer (thus assumes independence) and compares the relative increase of lower bit-widths wrt the 8 bit error. They combine their search algorithm with BRECQ to improve the overall quantization accuracy and claim their search can be combined with any quantization algorithm. The authors evaluate MixQuant for Resnets and MobileNet v2 on ImageNet.

**Summary Of The Review:**

The paper has major flaws in the empirical evaluation and also some mistakes in the proposed algorithm. Given the idea is simple and not novel, the paper clearly does not meet the expected quality for ICLR.

---

### Decision · Program_Chairs · 2023-01-20

**Decision:**

Reject

**Justification For Why Not Higher Score:**

This is a clear-cut case. All reviewers voted reject, and the authors did not submit a response.

**Justification For Why Not Lower Score:**

N/A

**Metareview: Summary, Strengths And Weaknesses:**

Summary:
This paper suggests a simple mixed precision search algorithm which evaluates the quantization error per layer and compare its relative increase in lower bit-widths in comparison with 8 bit error. They combine their search algorithm with BRECQ to improve the overall quantization accuracy, and evaluate it for Resnets and MobileNet v2 on ImageNet.

Strengths:
1) The search algorithm is very simple and fast.
2) Can be combined with any PTQ method.
3) The paper is clear.

As for weaknesses, there are many, here are a few examples:
1) Issues with the correctness of the algorithm.
2) Over-claiming the method is optimal.
3) Evaluation issues, e.g. comparisons not clear, some results make little sense.
4) No comparison with relevant previous papers, and limited novelty compared to these. For example, the Adaquant paper also had a mixed precision algorithm (in contrast to what has been claimed in this paper).
5) Only weights are quantized, not activations.
6) Limited hardware support to some configurations.

All reviewers voted reject, and the authors did not submit a response.